



# Response of particle number concentrations to Clean Air Action: Lessons from the first long-term aerosol measurements in a typical urban valley, West China

Suping Zhao [a,b,c], Ye Yu [a,b], Jianglin Li [a,b], Daiying Yin [d,e], Shaofeng Qi [a,e], Dahe Qin [c]

[a] Key Laboratory of Land Surface Process and Climate Change in Cold and Arid Regions, Northwest Institute of Eco-Environment and Resources, Chinese Academy of Sciences, Lanzhou 730000, China

[b] Pingliang Land Surface Process & Severe Weather Research Station, Pingliang, 744015, China

[c] State Key Laboratory of Cryospheric Science, Northwest Institute of Eco-Environment and Resources, Chinese Academy of Sciences, Lanzhou 730000, China

[d] Key Laboratory of Desert and Desertification, Northwest Institute of Eco-Environment and Resources, Chinese Academy of Sciences, Lanzhou 730000, China

[e] University of Chinese Academy of Sciences, Beijing 100049, China

*Correspondence to*: Suping Zhao (zhaosp@lzb.ac.cn); Ye Yu (yyu@lzb.ac.cn)

**Abstract.** The strictest ever Clean Air Action (CAA) has been implemented by Chinese government since 2013 to alleviate the severe haze pollution. The $PM_{2.5}$ mass concentration was found to be largely reduced in response to emission mitigation policies, but response of particle number concentrations (PNCs) to CAA was less evaluated in the previous studies, which may be largely different from $PM_{2.5}$ mass due to newly formed particle impacts. In this work, the first *in-situ* observation of particle number size distributions (PNSDs) during 2012-2019 in urban Lanzhou was used to analyze long-term PNCs variations and CAA impacts. The average number of particles in nucleation ($N_{13-25}$, particle number in the size range of 13-25 nm), Aitken ($N_{25-100}$, particle number in the size range of 25-100 nm) and accumulation ($N_{100-800}$, particle number in the size range of 100-800 nm) modes were respectively 2514.0 $cm^{-3}$, 10768.7 $cm^{-3}$, and 3258.4 $cm^{-3}$, and $N_{25-100}$ accounted for about 65.1% of total PNCs during the campaign. K-means clustering technique was used to classify the hourly mean PNSDs into six clusters, and each cluster corresponded to a specific source and influencing factor. The polluted clusters governed the winter PNCs before 2016, and their occurrence was less and less frequent after 2016, which was largely dominated by reduction in primary emissions. However, the contribution of new particle formation (NPF) events to summer $N_{13-25}$ decreased from 50% to about 10% during 2013 to 2015, and then increased to reach around 60% in 2019. The trends of size-resolved PNCs for each cluster were quantified by Theil-Sen regression. The size-segregated PNCs exhibited downward trends for all clusters during 2012-2015, especially in spring. The annual relative slopes of spring PNCs varied





from -54.7% to -17.2%, -42.6% to -14.1%, and -40.7% to -17.5% per year for 13-25, 25-100, and

100-800 nm size ranges, and the reduction in the polluted clusters was much larger than NPF clusters.

The ultrafine particle number was increased and the amplitude was much greater during 2016-2019.

The annual relative slopes of $N_{13-25}$ varied between 8.0% in fall and 135.5% in spring for NPF cluster.

5    In response to CAA, the increased daytime net radiation, higher ambient temperature and lower relative

humidity at noon for NPF events also could partly explain the higher $N_{13-25}$ induced by the more

frequent nucleation events after 2016, especially in spring. The air mass were mainly from the adjacent

regions of urban Lanzhou and less affected by long-range transport for NPF events, and the thus

particles were not easily grown by coagulation during transport processes, which was helpful for

10    occurrence of NPF events. Therefore, some effective control measures cooperatively controlled particle

number and mass should be took for the Chinese megacities.



## 1 Introduction

China has been experienced large-scale and long-lasting winter haze pollution due to fast-growing economy and urbanization in past decades. The high concentration of aerosols perturb radiative balance of the atmosphere and surface by directly scattering and absorbing solar radiation, or indirectly alter

cloud optical properties and lifetime serving as condensation nuclei and ice nuclei (Andreae and Rosenfeld, 2008; Gao et al., 2015; Li et al., 2017). The adverse effect of deteriorated air quality on public health is of the greatest concern in China (Hu et al., 2017; Lelieveld et al., 2015). The present air quality standards consider particle mass instead of number concentration (WHO, 2000). However, compared with the larger aerosol particles, the ultrafine particles (UFPs, diameter < 100 nm) scarcely

contribute to aerosol mass, while they share the largest number fraction in urban areas (Hussein et al., 2004; Wehner et al., 2004). The toxicity of UFPs is enhanced by the large surface area due to high number concentrations, and they can penetrate deep into lungs, ending up in the blood circulation (Oberdörster et al., 2005; Schmid and Stoeger, 2016).

Aerosol ability to efficiently scatter or absorb light depends not only on their chemical composition but on their sizes as well (Asmi et al., 2013). Liu et al. (2020) indicated that coating plays an important role in light absorption. The amplification of black carbon absorption by the coating increased from 1.21 to 1.75 with increasing aerodynamic diameter ($D_{ae}$) due to the thicker coating of BC-containing particles with a larger $D_{ae}$. Their study highlights the strong dependence of the microphysical and optical

properties of BC on size. The more recent study of Zhao et al. (2021) found that interdecadal AOD was negative trend from 2009 to 2018, which may be related to the variation in particle size distribution. Some aerosol monitoring networks were established around the world for long-term measurements of climate-relevant aerosol properties, such as Geophysical Monitoring for Climate Change (GMCC, Bodhaine, 1983) and Global Atmospheric Watch (GAW, Rose et al., 2021). The particle number

concentration (PNCs) and size distributions (PNSDs), considered to be a primary indicator of human impacts on atmospheric composition, was the main aerosol property measured at the networks. Based on long-term *in-situ* measurements at the network sites, many studies on particle number and size distributions have been conducted since the 1990s in Europe and North America (Asmi et al., 2011; Birmili et al., 2016; Dal Maso et al., 2008; Heintzenberg et al., 2011; Krecl et al., 2017; Kulmala et al.,

2004; Makela et al., 1997; Sun et al., 2020; Wiedensohler et al., 2012). Their studies indicated that the



annual, weekly and diurnal cycles largely depended on station type and geographic location. The more recent study of Schmale et al. (2018) also well illustrated the importance of measuring the PNSD over long enough time periods in contrasting micro-environments for the understanding of aerosol-climate interactions and the improvement of their representation in numerical models. Sun et al. (2020)

determined long-term trends of PNCs during 2009-2018 for 16 sites ranging from roadside to high Alpine environments, and the annual relative slope varied from -17.2% to -1.7%, -7.8% to -1.1%, and -11.1% to -1.2% per year for 10-30 nm, 30-200 nm and 200-800 nm size bins, respectively. The downward trends of PNCs were found to be dominated by the reductions in various anthropogenic emissions, while meteorology impacts were less important or negligible. However, a few long-term

PNSD measurements in the developing countries mainly concentrated on urban Beijing since 2004 (Wang et al., 2013; Wehner et al., 2004), the North China Plain since 2008 (Shen et al., 2011), and Mount Waliguan since 2005 (Kivekas et al., 2009). Aitken mode particles (25-100 nm) were found to be accounted for about half of total PNCs in urban areas in China (Wu et al., 2008), and number in accumulation mode (100-1000 nm) was around 4 times higher than that in the developed countries

(Wehner et al., 2008; Wu et al., 2008), indicating that largely different PNSD characteristics in China from Europe and North America.

PM$_{2.5}$ (particulate matter with aerodynamic diameter less than 2.5 μm) decreased by 30%-50% across China over the 2013-2018 period in response to Air Pollution Prevention and Control Action Plan

(APPCAP) in 2013 implemented by Chinese central government (Zhai et al., 2019). Compared with PM$_{2.5}$ mass concentrations, the particle number concentrations were more directly affected by newly formed particles (Dal Maso et al., 2008; Dinoi et al., 2021), and new particle formation (NPF) events contributed about 54% of total PNCs in Leipzig, Germany (Ma and Birmili, 2015). Guo et al. (2014) tried to reveal the causal connection between NPF and haze pollution, and reported that NPF trends to

precede winter haze episodes in Beijing. The more recent study by Kulmala et al. (2021) found that over 65% of the number concentrations of haze particles resulted from NPF events in Beijing, and their findings suggested that almost all present-day haze episodes originated from NPF, mainly since primary emission considerably decreased during recent year. PNSDs were considered to be better indicators of the strength of emission sources (Vu et al., 2015), but they were more easily modulated by

aerosol dynamic processes, such as nucleation, coagulation, volatilisation and condensations (Birmili et



al., 2010; Kulmala, 2003). Nucleation and coagulation were largely affected by coagulation sink (CoagS), and CoagS significantly decreased due to large reduction of $PM_{2.5}$ mass concentrations in response to APPCAP. Therefore, response of particle number concentration in different size bins to emission mitigation policies may be different from $PM_{2.5}$ mass concentration.

The long-term PNSDs measurements were mainly conducted before APPCAP in China, and it has been less reported that the response of particle number concentrations to the strictest ever air pollution control policies implemented by Chinese central government. Lanzhou, as one of the most polluted cities around the world at special basin terrain, obtained Today's Transformative Step 2015 awarded by

the United Nations due to significant improvement in urban air quality (Zhao et al., 2018). The atmospheric horizontal and vertical dispersion conditions inside the basin are poor due to weak winds and strong multi-layer temperature inversion induced by basin terrain (Pandolfi et al., 2014). Therefore, the air pollutants were easily trapped inside the basin and hard to disperse to the upper air. Furthermore, basin aerosol pollution was largely controlled by vertical than horizontal dispersion as compared to the

plain (Zhao et al., 2019). Based on a unique PNSD dataset for the period of 2012-2019 at urban Lanzhou in West China, this study investigates the long-term trends of PNCs in different modes, to evaluate the role of emission reduction and meteorology in PNC variations. The results of this study may be important for the policymakers to cooperatively prevent and control heavy particle mass and number concentrations in Chinese megacities.

## 2 Data and methods

### 2.1 Measurement site descriptions

Lanzhou, located at the intersection of Tibetan Plateau, Losses Plateau and Mongolia Plateau, is in a long valley running from the east to the west. The urban area is encircled by the hills rising from 200 m

to 600 m, and thus formed saddle-shaped basin terrain (Figure 1). The weak winds and multi-layer temperature inversion occurred frequently due to terrain impacts, and thus the air pollutants are trapped inside the basin (Chu et al., 2008). It was thought to be one of the most polluted cities around the world (WHO, 2014), and photochemical smog episode (PSE) was observed in the 1980s at Xigu District of urban Lanzhou, which was the first time PSE was observed in China (Chen et al., 1986). The

observation campaign was conducted from September 2012 to August 2019 on the rooftop of a 32-m


high research building of the Northwest Institute of Eco-Environment and Resources (NIEER), Chinese

Academy of Sciences. There are two major roads with traffic volume more than 2000 cars per hour

near the observation site (Figure 1). The NIEER is surrounded by residential and commercial buildings,

and there are no local industrial sources around the site (Zhao et al., 2015a), and thus the measurement

site can represent urban background.

### 2.2 PNSD, criteria air pollutants and meteorology data

5-min particle number concentrations and size distributions (13-800 nm) were measured continuously

by scanning mobility particle sizer (Model 3936, TSI, USA) for about 7 years at the urban site from

September 2012 to August 2019. The aerosol and sheath flowrates were set to 0.3 L min$^{-1}$ and 3 L min$^{-1}$,

respectively. The sampling inlet was mounted 1.5 m above the rooftop. The diffusional and

gravitational losses for the inlet lines of SMPS were calibrated during the campaign. The SMPS's

mobility was calibrated with monodisperse aerosols prior to their deployment in the field. The impactor

was cleaned every day and aerosol and sheath rates were examined with a bubble flow meter to insure

the good performance of the instrument. The each PNSD was parameterized with a least-square

log-normal fitting method providing parameters of 2-3 log-normal modes (Birmili et al., 2001). Three

modes ($i$ = 1, 2, 3) were used corresponding to the nucleation mode (13-25 nm), Aitken mode (25-100

nm) and accumulation mode (100-800 nm), respectively. The log-normal distribution is expressed as

(Seinfeld and Pandis, 2006):

$$\frac{dN}{d \log D_p} = \sum_{i=1}^{n} \frac{N_i}{\sqrt{2\pi} \log \sigma_i} \exp\left(-\frac{\left(\log D_p - \log \overline{D}_{p,i}\right)^2}{2\left(\log \sigma_i\right)^2}\right) \qquad (1)$$

where $N_i$ is the total number concentration of the mode $i$, $\overline{D}_{p,i}$ is the median diameter of mode $i$, $\sigma_i$ is

the geometric mean standard deviation of the distribution and $n$ is the number of the modes. The

symbol of log means log10 in this study.

The hourly averaged concentrations of the criteria air pollutants (PM$_{2.5}$, PM$_{10}$, SO$_2$, NO$_2$, O$_3$, CO) were

measured at Lanzhou Biologicals Institute, which is around 2.8 km away from the observation site. SO$_2$,

NO$_2$, CO and O$_3$ are measured by the ultraviolet fluorescence method, the chemiluminescence method,

the non-dispersive infrared absorption method and the UV-spectrophotometry method, respectively.





PM$_{2.5}$ and PM$_{10}$ are measured by micro oscillating balance method. Aerosol optical parameters (AOD and Alpha) were measured continuously by a sun tracking spectrophotometer (CE-318, CIMEL, France) during the campaign. The 10-min meteorological parameters including temperature, relative humidity, wind speed and direction, precipitation and raindrop size distribution, and solar radiation were

monitored by an automatic meteorological station co-located with the observation site. All the on-line data were hourly averaged and presented at local time (Beijing Time = UTC+8) throughout this paper.

**2.3 Trend analysis methods**

Referring to the method used in the study of Sun et al. (2020), a customized Sen-Theil trend estimator

was used to analyze the long-term trends of PNCs in nucleation, Aitken, and accumulation modes, the concentrations of the criteria air pollutants and meteorological parameters in this study. The technique can calculate the true slope of the parameters by considering the impact of their seasonal, weekly and diurnal cycles, and avoid the effect of outliers and missing values. The change rates for the hourly or daily time series can be calculated with the below equation:

$$m_{i,k} = \frac{\left(x\left(i+\Delta t\right) - x\left(i\right)\right)}{\Delta t} \tag{2}$$

Where $k$ is the integer. $\Delta t$ is equal to the product of $k$ and 364 days (52 weeks), indicating that data points from two different years are compared only if they were measured in the same hour of the day, day of the week, and season the year.

**2.4 Cluster analysis methods**

To extract some more valuable information, K-means clustering method used in various studies has been considered to be a preferred technique for data analysis in environmental fields (Sabaliauskas et al., 2013; Tunved et al., 2004). The K-means clustering routine split the multi-dimensional data into predefined number of subgroups, and clusters are as different as possible from each other, but as

homogeneous as possible within themselves, by iteratively minimizing the sum of squared Euclidean distances from each member to its cluster centroid. Cluster analysis was used to divide hourly mean PNSDs during the campaign into several groups with comparable particle number in different size bins within groups. The K-means clustering algorithm available in MATLAB$^{©}$ was used in this study. Based on the rule with maximum inter-cluster and minimum intra-cluster variances, the determination of



number of clusters, a very complicated problem, was conducted by statistical software SAS© in this study, which was given in detail in the study of Zhao et al. (2016).

### 3 Results and discussion

A continuous 7.5 years dataset was evaluated in this investigation. Except the instrument maintenance and re-location, 80% of the data was effective. The continuous PNSD dataset was integrated to calculate PNCs in different size bins. In this study, diameter ranges for the nucleation mode, Aitken mode, and accumulation mode were determined as 13-25 nm, 25-100 nm, and 100-800 nm, respectively (Dal Maso et al., 2005). The total PNCs covered from 13 nm to 800 nm in mobility diameter.

### 3.1 Overview of the particle number concentration

Sources and origins of particles in the three modes may be largely varied at a specific micro-environments. Nucleation mode particles are from atmospheric nucleation events which is closely related to the low volatile condensable gases such as water and sulfuric acid and growth of the smaller aerosol particles (Kulmala, 2003). Aitken mode particles are primarily emitted from combustion processes, such as coal combustion for domestic heating in wintertime, and also from hygroscopic growth and coagulation of nucleation mode particles. For relatively clean environment, the growth of nucleation mode particles is predominant due to less coagulation sink (Rose et al., 2021), while the primary emissions are more important at the highly polluted urban areas (Hussein et al., 2004). Accumulation mode particles originate from coagulation and hygroscopic growth of Aitken mode particles and long-range transport from the highly polluted areas.

Figure 2 shows variation of particle number in nucleation mode ($N_{13-25}$), Aitken mode ($N_{25-100}$), and accumulation mode ($N_{100-800}$), aerosol optical properties (AOD, Alpha), criteria air pollutants (PM$_{2.5}$, O$_3$, SO$_2$, and NO$_2$) and basic meteorological parameters (wind speed, relative humidity, temperature, net radiation) during the entire measurement campaign. The probability density functions and the corresponding statistical parameters also were given in Figure 2. The mean values for $N_{13-25}$, $N_{25-100}$, and $N_{100-800}$ were 2514.0 cm$^{-3}$, 10768.7 cm$^{-3}$, and 3258.4 cm$^{-3}$, respectively. Aitken mode particles, accounting for 65.1% of total PNCs, were significantly higher than the other modes, and the differences



were much larger than the results at European cities (Cusack et al., 2013; Leoni et al., 2018), urban

Beijing (Wu et al., 2008) and North China Plain (Shen et al., 2011), which may be related with the fact

that the particles in 3-12 nm and 800-1000 nm were not covered in nucleation and accumulation mode

for our measurement campaign. The nucleation mode particles also can grow to Aitken mode by

coagulation and hygroscopicity when they were transported from the primary emissions at the ground

surface to the sampling location due to the much higher sampling height compared to the other studies.

The mean $PM_{2.5}$ and $O_3$, $SO_2$, $NO_2$ concentrations were 49.9 μg m$^{-3}$, 44.8 μg m$^{-3}$, 22.7 μg m$^{-3}$ and 57.5

μg m$^{-3}$ during 2014-2019, and mean values of wind speed, temperature, relative humidity, and net

radiation were 1.6 m s$^{-1}$, 10.9 ℃, and 44.6%, and 44.4 W m$^{-2}$, respectively.

Until now, numerous measurements of sub-micron PNSDs have been carried out at a variety of

locations to examine their variations and key influencing factors. Table 1 summarizes experimentally

determined particle number concentrations in the troposphere for the measurement campaigns

conducted longer than 1 year across the globe. The mean number concentrations in the three modes

were much lower than that in urban Beijing (Wang et al., 2013), and significantly higher than that at a

remote background station, Mt. Waliguan, one of global GAW sites in China (Kivekas et al., 2009). The

sub-micron particle number concentration was much lower compared to the most polluted cities in

India, such as Delhi (Gani et al., 2020) and Kanpur (Kanawade et al., 2014), especially Aitken and

accumulation modes. Number of the particles in Aitken and accumulation modes at Asian cities were

even higher than that at the urban site in Europe and North America, which may be largely related to

poor visibility at Asian cities according to Mie scattering theory. For nucleation mode, the situation is

opposite, which may be because newly formed particles rapidly scavenged by coagulation with a

mount of the larger particles at highly polluted cities (von Bismarck-Osten et al., 2013; Wang et al.,

2011).

### 3.2 Trends of PNCs, criteria air pollutants, and meteorological parameters

Besides primary emissions from human activities at urban areas, particle number concentration was

easily affected by secondarily new-formed particles, which was closely related to meteorological

conditions such as temperature, relative humidity and new radiation (Zhao et al., 2015a). Figure 3

shows inter-annual variations of monthly averaged particle numbers, criteria air pollutants and wind





speed during 2012-2019, and normalizes the time series data ($N_{13-25}$, $N_{25-100}$, $N_{100-800}$, PM$_{2.5}$, O$_3$, SO$_2$, NO$_2$, and wind speed) to fix values to equal 100 at the beginning of September 2012. The particle number in the three size ranges declined largely during 2012-2015 (Period I), and summer $N_{13-25}$ decreased by around 75% in 2015 compared to that in 2013, while that in winter less varied during

Period I. The $N_{25-100}$ and $N_{100-800}$ reduced more in winter than that in summer due to emission control impacts. The number in nucleation mode particles ($N_{13-25}$) increased significantly during 2016-2019 (Period II), which was consistent with O$_3$ while showed the opposite trend with declining PM$_{2.5}$ during Period II. The strongly declining aerosol radiative effect due to the strict air pollution controls resulted in an unprecedented rapid increasing trend in surface solar radiation over China during 2014-2019 (Shi

et al., 2021), which maybe promote formation of secondary air pollutants.

The particle number in the Aitken and accumulation modes ($N_{25-100}$, $N_{100-800}$) firstly increased during 2016-2017 and then decreased from 2018 to 2019, and their variations were consistent with the primary emitted pollutants (SO$_2$, NO$_2$), indicating that $N_{25-100}$ and $N_{100-800}$ variations during 2016-2018 were

mainly modulated by primary emissions. Sun et al. (2020) analyzed the long-term trends of particle number concentrations at 16 observational sites in Germany from 2009 to 2018, and number concentrations of particles in the three modes were found to be significant decreasing trends in response to emission mitigation policies. The contrasting response of nucleation mode particle to mitigation policies between China and Germany may be related to the fact that more reduction of

coagulation sink due to the ever strictest Clean Air Action in China, and thus NPF event easily occurred due to less coagulation scavenging effects (Gani et al., 2020). The variation in wind speed was not significant during the entire measurement campaign.

In view of the contrasting PNC trends between Periods I and II, the below analyses compared mean

diurnal and annual variations of particle number in the three size bins ($N_{13-25}$, $N_{25-100}$, and $N_{100-800}$), PM$_{2.5}$ and O$_3$ as wind directions before and after January 2016 (Figures 4, S1-S4). The most obvious increase in $N_{13-25}$ was during 12:00-16:00 in summer months after January 2016 compared to before January 2016, and the largest increase corresponded to easterly, southerly and southwesterly winds, especially for the annual cycles with the more significantly increased $N_{13-25}$ for southeasterly winds.

The large CPF values of $N_{13-25}$ mainly corresponded to southerly winds (Figure 5), which can support





the above results. $N_{25-100}$ difference between the two periods (2012-2015 vs. 2016-2019) was much less

significant than $N_{13-25}$, and the most obvious $N_{25-100}$ increase occurred in morning and evening rush

hours for northeasterly winds (Figure S1), which could be supported by the results of polar plots

(Figure 5). Figure 6 illustrated mean particle number size distributions by varying wind directions, and

number in Aiken mode particles for north northeasterly winds ($0°$ - $45°$) was largely higher than that for

the other wind directions. The emissions from jammed traffic and winter domestic heating with

traditional stoves at Keji Street, about 500 m away from the sampling site, could be transported to the

station as northeasterly winds. In addition, the larger increase in $N_{25-100}$ at 13 p.m. for easterly, southerly

and southwesterly winds was consistent with $N_{13-25}$ possibly due to new-formed particle growth

impacts.

The $N_{100-800}$ and PM$_{2.5}$ trends from Period I to II in diurnal and annual cycles were opposite to $N_{13-25}$

with the significant reduction at noon in summer months for southerly winds (Figures 5, S2 and S3),

which was mainly affected by Clean Air Action (Li et al., 2021). Gani et al. (2020) studied particle

number concentrations and size distributions in a polluted megacities of Delhi in India, and pointed out

that strategies that only target accumulation mode particles (which constitute much of the fine PM$_{2.5}$

mass) may even lead to an increase in the UFP concentrations as the coagulation sink decreases.

Furthermore, O$_3$ increased more significantly in the afternoon in summer months after than before

January 2016, and wind directions for the largest increased O$_3$ concentrations were consistent with

nucleation mode particles (Figure S4), which further confirmed that the increased $N_{13-25}$ from Period I

to II was induced by more frequent nucleation events. Compared with before January 2016, the more

favorable meteorological conditions after January 2016 such as the much drier air (Figure S5), higher

ambient temperature (Figure S5) and stronger solar radiation (Figure S6) for southerly winds also

helped to form new particles, which could be supported by our previous work in the same site (Zhao et

al., 2015a).

### 3.3 Typical particle number size distributions influenced by varying factors

Besides chemical composition of airborne particles, the information derived from particle number size

distributions (PNSDs) is beginning to play an important role in source apportionment studies (Vu et al.,

2015) due to the obvious difference of diameters for the particles from varying sources. The hourly





average PNSDs during the entire measurement campaign were classified into six clusters by K-means clustering technique, and mean PNSD for each typical type was showed in Figure 7. As showed in Figure 7, the shape and mode diameter of PNSDs were largely different among the Clusters. Mode diameters varied from ~ 20 nm for Cluster B to 70 nm for Cluster F, and more than a quarter of PNSDs

was sorted into Cluster A with mode diameter of ~ 55 nm, while Cluster B less occurred with mode diameter of ~ 20 nm. The sources and key factors influencing each cluster of PNSD can be better determined by averagely annual and diurnal variations of occurrence frequencies, and the corresponding air pollutants and meteorological parameters for the Clusters (Figure 8, Tables 2 and 3).

About 70% of Clusters A and F is in the cold seasons (October-December, and January-March) with the almost opposite diurnal pattern between the two Clusters (Figure 8). Clusters A and F had the highest number concentrations of accumulation mode particles ($N_{100-800}$) and mass concentrations of particulate matter ($PM_{2.5}$, $PM_{10}$) and gaseous pollutants ($SO_2$, $NO_2$, $CO$), while the lowest particle number in nucleation mode ($N_{13-25}$) and $O_3$ mass concentrations among the clusters (Table 2), suggesting that the

two polluted clusters may be mainly impacted by primary emissions from human activities, and was defined and abbreviated as "Pollut_C" in the below analyses. That also can be confirmed by larger geometric median diameters for the three modes ($GMD_{nuc}$, $GMD_{Ait}$, and $GMD_{acc}$) than that for the other clusters, and the high particle number concentrations in morning and evening rush hours (Figure S7). Compared with other clusters, the weaker winds and net radiation, lower ambient temperature, and

higher relative humidity indicated that the severe air pollution for Clusters A and F was significantly affected by stable air and poor diffusion conditions. Furthermore, as illustrated in Figure 9, Cluster F (A) accounted for more than 40% (60%) of all clusters during five hours after occurrence of Cluster A (F), and the more frequently synchronous occurrence between Clusters A and F maybe related to the pollution process from "light-severe-light" episodes. From the perspectives of inter-annual variations in

occurrence frequency, the Pollut_C increasingly less occurred from 2014 to 2019 possibly due to implementation of Clean Air Action (Figure 8A), which will be analyzed in detail in the following section.

The Clusters B and E mainly appeared in the daytime in the warmer months, and occurrence frequency

had a sharp peak in the afternoon (Figures 8B and 8C), and the peak for Cluster E lagged around two



hours than that for Cluster B. The frequency of Cluster E during two hours after occurrence of Cluster B was larger than 80%, and mode diameter of Cluster E (~ 30 nm) was only larger than that of Cluster B (~ 20 nm), and thus it was inferred that Cluster B represented secondarily new particle formation (NPF) event impacts, while Cluster E was influenced by subsequently new particle growth events. The

inference could be confirmed by the highest particle number in nucleation mode and $O_3$ mass concentration among the clusters (Table 2). The sharply increased nucleation mode particles at 9:00 was followed by a subsequent growth to accumulation mode indicated by the typical "banana-shaped" temporal development of the number size distribution (Figure S7, Boy and Kulmala, 2002), which also supported the above inference. In addition, the less coagulation sink such as low number concentrations

of particles in accumulation mode, and low $PM_{2.5}$ and $PM_{10}$ mass induced by higher wind speed helped to form secondarily new particles (Tables 2 and 3). The more recent study of Gani et al. (2020) investigated particle number concentrations and size distribution in a polluted megacity: the Delhi, and found that reduction in mass concentration in the highly polluted megacity may not produce a proportional reduction in PNCs, and may even lead to an increase in the UFP concentrations as the

coagulation sink decreases. The mean AOD of 0.39 for Cluster B was significantly lower than that for the other clusters (Table 2), which resulted in the higher atmospheric transparency and thus stronger net radiation (223.55 W m$^{-2}$) and higher ambient temperature (20.77 °C). The drier air was conducive to detecting NPF events and newly formed particles grown hard by hygroscopic growth under low RH environments. The occurrence frequency for the two clusters first reduced from 2013 to 2015 and then

increased until 2019, which contrasted with the "Pollut_C" during the campaign. Clusters B and E were abbreviated as "NPF_C" for the following analyses.

The mean PNSD for Cluster C was much wider and more flat than that for the other clusters and thus it was hard to determine the mode diameter, especially for the PNSDs from dawn to noon (Figures 7 and

S7). The number of particles in nucleation mode ($N_{13-25}$) was only lower than NPF_C while that in accumulation mode ($N_{100-800}$) was only lower than Pollut_C. The cluster was more easily occurred in the morning in the warm months, which was consistent with most of the clusters except the polluted Clusters A and F. Additionally, except for Cluster F, the occurrence frequency of the other clusters was comparable and ranged from about 15% to 28% during 1-12 hours after Cluster C. The frequency of

Cluster C also less varied during 1-12 hours after Clusters A, D, E and F (Figure 9). Combined the



above fact with the modest concentrations of particle number in the three modes, criteria air pollutants and meteorological parameters, it was inferred Cluster C representing urban background PNSD, and thus it was defined as "UB_C" in the following analyses. Cluster D more frequently occurred in the morning and evening rush hours in the warm seasons (Figure 8), and the correspondingly mean particle

number in Aitken mode ($N_{25\text{-}100}$) was the second highest ever-just behind Cluster A, which may be impacted by motor vehicle emissions from the nearby roads. The mode diameter of ~ 40 nm was only larger than NPF_C (Clusters A and E), and it appeared frequently after Cluster E with the high concentration of particles in Aitken mode in the afternoon (Figure S7) possibly due to new particle growth impacts. Therefore, Cluster D was jointly influenced by motor vehicle emissions and NPF

events, and was defined as "VE_NPF_C" in the following section.

From the perspectives of the variation in mode diameter among the clusters (Figure 7) and the variation in frequency during 1-12 hours after each cluster (Figure 9), the NPF_C was closely followed by Pollut_C during the measurement campaign, and the clusters can be ranked by temporal order as

"B→E→D→A→F". Therefore, NPF events significantly contributed to haze episodes in the subsequent 1–2 days, which may be increasingly obvious mainly due to considerably decreased emissions of primary particles during recent years in response to Clean Air Action. Guo et al. (2014) first reported that atmospheric NPF tends to precede winter haze episodes in Beijing, and then the latest study of Kulmala et al. (2021) investigate how NPF and subsequent particle growth affect the initial

steps of haze formation in Beijing. Their findings showed that reducing the subsequent growth rate of freshly formed particles by a factor of 3–5 would delay the buildup of haze episodes by 1–3 days.

**3.4 Impact of Clean Air Action on PNC variations**

The response of PM$_{2.5}$ mass to Clean Air Action has been evaluated in many previous studies, and

PM$_{2.5}$ was found to be decreased by 30%-50% across China during 2013-2018 due to the implementation of emission control policies (Zhai et al., 2019). The impact of the policies on particle number may be more complex as compared to PM mass since more fine particles cannot rapidly grow by coagulation with the reduced coarse particles (Gani et al., 2020). However, the response of PNCs to the restricted emissions was only analyzed by some short-term measurements during some important

and international meetings and activities such as the summer Olympic Games in 2008 and the



Asia-Pacific Economic Cooperation (APEC) in 2014 and China's V-Day parade in 2015 (Chen et al.,
2015; Shen et al., 2016; Wang et al., 2013). The Long-term *in-situ* measurements of PNSDs and mass
concentrations of the criteria air pollutants was essential to understand the emission control impacts and
to reveal the mechanism. Figures 10, S8 and S9 show trends of daily mean particle number in the three

modes as wind directions for each cluster based on 7.5 years measurement. The number of particles in
nucleation mode ($N_{13-25}$) first decreased from 2012 to 2015, and then increased rapidly after 2016. The
$N_{13-25}$ changing trend for NPF_C (Clusters B and E) was more significant as compared with that for the
other clusters, especially for southeasterly winds. The specific winds corresponded to more $PM_{2.5}$
reduction on summer afternoon after than before 2016 due to impact of emission mitigation policies

(Figure S3), and thus NPF events, represented by NPF_C, were easily detected by chemical reactions
due to reduced coagulation sink. The more solar radiation reached near-surface air as a result of
reduced $PM_{2.5}$ mass (Shi et al., 2021), and thus ambient temperature increased and relative humidity
declined (Figures S5 and S6), which also favored occurrence of NPF events (Zhao et al., 2015a).

At our sampling site, $N_{25-100}$ was easily influenced by growth of newly formed particles and primary
emissions from human activities. The $N_{25-100}$ trends was similar with $N_{13-25}$ for Clusters B, D and E, and
the increasing trends also were more significant after 2016 for southwesterly winds (Figure S8), which
represented NPF impacts. Dependence of $N_{25-100}$ on wind directions was not obvious for Clusters A, C
and F, and thus the trends may be related to variations in primary emissions. Unlike nucleation and

Aitken modes, particle number in accumulation mode ($N_{100-800}$) less depended on wind directions.
Furthermore, $N_{100-800}$ was the lowest and less varied for NPF_C, while that for Clusters A, C and D had
similar trend with $N_{25-100}$, and that for Cluster F, the most polluted cluster, was downward trend during
the campaign due to implementation of the strictest ever Clean Air Action. Therefore, the response of
particle number to air pollution control may be largely different for each size fraction, which may be

closely related to the variations in coagulation sink and meteorological conditions induced by reduced
primary emissions. That will be discussed in detail in the following section.

To better evaluate variations in particle number concentrations and emission control impacts, Figures
11, S10 and S11 show variations of contributions of each cluster to monthly averaged PNCs in 13-25

nm, 25-100 nm and 100-800 nm during the campaign, respectively. Pollut_C (Clusters A and F)



dominated the winter PNCs in different size bins before 2016, and their occurrence was less and less

frequent after 2016, especially for the most polluted Cluster F, which was largely dominated by

reduction in primary emissions. Contrasting to Pollut_C, as a main cluster representing NPF events, the

contribution of Cluster B to summer $N_{13\text{-}25}$ decreased from 50% to about 10% during 2013 to 2015, and

then increased to reach around 60% in 2019. For Cluster C representing urban background, its

frequency less varied during the entire measurement campaign. The particle number was dominated by

primary emissions before 2016, and thereafter that was controlled by NPF events, which was partly due

to emission control. In response to air pollution control, the reduction in coarse particles could promote

secondary new particle formation by reduced coagulation sink (Gani et al., 2020). NPF events were

largely dependent on PM mass concentrations mainly contributed by coarse aerosol particles.

Accumulation mode particle number concentrations in cities of developing countries are generally

higher than that in many western cities (Gani et al., 2020; Wu et al., 2008), and thus the response of

NPF events to emission control may be largely different between the cities of developed and

developing countries. For example, Sun et al. (2020) found that coincidently downward trends of

particle number and black carbon mass concentrations at 16 observational sites in Germany from 2009

to 2018 due to reduced anthropogenic emissions. Gani et al. (2020) pointed out that strategies that only

target accumulation mode particles in a polluted megacity in India may even lead to an increase in the

UFP concentrations as the coagulation sink decreases. Shen et al. (2016) also found that $PM_1$ mass

concentration was significantly reduced while NPF event frequency was much higher during short-term

emission control period.

We also quantitatively evaluated the changing trends of particle number in the three modes by

Theil-Sen regression. In view of the contrasting trends, the observation period was divided into two

sub-periods, i.e., before and after January 2016. Figures 12 and 13 illustrated seasonal and diurnal

variations of trends of PNCs for each cluster during each sub-period. For Period I (2012-2015), PNCs

in the three size bins exhibited downward trends for all clusters, especially in spring. The annual

relative slopes of spring PNCs varied from -54.7% for Cluster F to -17.2% for Cluster B, -42.6% for

Cluster A to -14.1% for Cluster B, and -40.7% for Cluster A to -17.5% for Cluster B per year for 13-25,

25-100, and 100-800 nm size ranges (Figure 12). The PNCs for Pollut_C (Clusters A and F) decreased

by about 40% in the morning and evening rush hours, which was much higher than that in the other





hours of day. Therefore, the more reduction in PNCs for Pollut_C may be closely related to emission

control policies. The much larger PNC reduction in this study than that in Germany may be due to

implementation of the stricter ever emission mitigation policies in Chinese cities (Sun et al., 2020).

Contrary to Period I, the UFPs number was increased and the amplitude was greater during Period II

(2016-2019). The annual relative slopes of $N_{13-25}$ varied between 5.1% (fall) and 314.4% (winter), 8.0

(fall) and 135.5% (spring), 11.3% (fall) and 184.3% (winter), -4.5 (fall) and 59.1% (summer), 6.3%

(fall) and 30.3% (spring), and 3.6% (fall) and 15.7% (spring) for Clusters A-F. The maximum increase

of $N_{13-25}$ was in the spring afternoon for NPF_C, which may be governed by NPF events due to reduced

coagulation sink corresponding to low $N_{100-800}$. The winter $N_{13-25}$ was increased significantly for Cluster

A during Period II, especially in the morning and evening rush hours, suggesting the impact of primary

emissions from motor vehicles. The annual slope of $N_{100-800}$ less varied as compared to that of ultrafine

particles in seasonal and diurnal cycles.

### 3.5 Role of meteorology and air masses

NPF events predominantly occurred under dry and sunny weather conditions (Birmili and

Wiedensohler, 2000). Figures 14 and 15 show the diurnal and inter-annual variations in meteorological

parameters such as net radiation, temperature, relative humidity, and wind speed and direction for each

cluster during the campaign to better understand response of PNCs to meteorology. The peak of net

radiation and $N_{13-25}$ coincided at noon for NPF_C, and their peaks were largely higher than that for the

other clusters (Figure 14). The increased daylight net radiation for Cluster B also could partly explain

the higher $N_{13-25}$ induced by the more frequent NPF events after 2016, especially in spring (Figures 15

and S12). The higher ambient temperature and lower relative humidity at noon and the larger daily

ranges for NPF_C also indicated that dry and hot air in sunny day was conducive to form new particles.

In addition, NPF events corresponding to large southeasterly winds may be because accumulation

mode particles were dispersed and diluted by strong winds and thus coagulation sink decreased, which

can be supported by the above results. According to empirically based mathematical function between

number concentrations of fine particles (FP, diameter <2.5 μm) and meteorological variables, Hussein

et al. (2006) found that the predicted number concentrations of accumulation mode particles follow this

relationship more closely than those of UFP's due to the origin and type of aerosol particles in the

accumulation mode size range, being mainly regional and long-range transported. The main limitation



of the mathematical function in their study was during NPF events, indicating that particles in
nucleation and accumulation modes were differently dependent on meteorological variables.

A general finding was that changes in aerosol were related to air mass changes (Birmili et al., 2001),

and dust aerosols from Gobi Deserts at Hexi Corridor could be transported to Lanzhou and affected
urban PM pollution (Zhao et al., 2015b). Figure 16 illustrates gridded back trajectory frequencies with
hexagonal binning for each cluster to explore the impacts of air mass on variations of particle number.
The huge discrepancy of back trajectory frequencies among the six clusters suggested that the air mass
history has a significant impact on urban particle number concentrations and size distributions. For

example, back trajectories were mainly from the adjacent regions of urban Lanzhou and less affected
by long-range transport for NPF_C, and the thus particles were not easily grown by coagulation during
transport processes, which was conducive to occurrence of NPF events. In urban Beijing, Wang et al.
(2013) also indicated that mean total PNCs from north directions were higher than the air masses that
came from other directions, while more volume concentrations were observed for the air masses from

the southwest and the south. Therefore, particle number size distributions in urban Lanzhou were partly
affected by air mass conditions.

**4 Summary and conclusions**

The first *in-situ* observations of particle number size distributions (PNSDs) in the size range of 13-800

nm were conducted from 2012 to 2019 in urban Lanzhou, a typical valley city in west China.
Meanwhile, the mass concentrations of the criteria air pollutants ($PM_{2.5}$, $PM_{10}$, $O_3$, $SO_2$, $NO_2$, and CO),
AOD and meteorological variables (temperature, relative humidity, wind speed and direction and net
radiation) also were measured during the campaign. The customized Sen-Theil trend estimator and
K-means clustering technique were used to explore the trends of PNCs and the criteria air pollutants,

and to reveal the contributions of variations in primary emissions due to Clean Air Action and
secondary formation to PNCs. Some novel findings were obtained as follows.

The mean values for particle number in nucleation ($N_{13-25}$), Aitken ($N_{25-100}$) and accumulation modes
($N_{100-800}$) were respectively 2514.0 $cm^{-3}$, 10768.7 $cm^{-3}$, and 3258.4 $cm^{-3}$, and $N_{25-100}$ accounted for about

65.1% of total PNCs during the campaign. The particle number in the three modes declined largely





during 2012-2015 such as summer $N_{13-25}$ decreased by around 75% in 2015 compared to that in 2013. However, $N_{13-25}$ increased significantly during 2016-2019, which was consistent with $O_3$ while showed the opposite trend with declining $PM_{2.5}$ during the period. The most obvious increase in $N_{13-25}$ was during 12:00-16:00 in summer months, and the largest increase corresponded to easterly, southerly and

southeasterly winds. The $N_{25-100}$ and $N_{100-800}$ firstly increased during 2016-2017 and then decreased until 2019, and their variations were consistent with the primary emitted pollutants ($SO_2$, $NO_2$). $N_{25-100}$ difference between the two periods (2012-2015 vs. 2016-2019) was much less significant than $N_{13-25}$, and the most obvious $N_{25-100}$ increase occurred in morning and evening rush hours for northeasterly winds. In diurnal and annual cycles, the $N_{100-800}$ and $PM_{2.5}$ trends for the two periods were opposite to

$N_{13-25}$ with the significant reduction at noon in summer months for southerly winds, and thus decreased coagulation sink was conducive to occurrence of NPF events.

K-means clustering technique was used to classify the hourly average PNSDs into six clusters during the measurement campaign. The shape and mode diameter of PNSDs were largely different among the

clusters with varying mode diameters from ~ 20 nm to 70 nm. According to the annual and diurnal variations of occurrence frequency, PNSD, the corresponding air pollutants and meteorological parameters, the sources and key influencing factors were determined for each cluster. The two most polluted clusters (A and F), Pollut_C, were mainly affected by the primary emissions from human activities and poor diffusion conditions. Cluster B was followed by Cluster E, and $N_{13-25}$ had a sharp

peak in the afternoon in the warm months, and thus the two clusters represented new particle formation and growth event impacts. Cluster C suggested urban background PNSD, while Cluster D was jointly affected by motor vehicle emissions and NPF events. The response of particle number to air pollution control was largely different for each size fraction, which may be closely related to the variations in coagulation sink and meteorological conditions induced by reduced primary emissions. Based on trends

of daily mean particle number in the three modes as wind directions for each cluster, the contributions of primary emissions and secondary formation to PNCs were evaluated in this study. The southeasterly winds corresponded to more $PM_{2.5}$ reduction on summer afternoon in response to emission control policies and thus more solar radiation reached ground surface, which promoted NPF occurrence due to decreased coagulation sink. The polluted clusters governed the winter PNCs before 2016, and their

occurrence was less and less frequent after 2016, which was largely dominated by reduction in primary


emissions. However, the contribution of NPF events to summer $N_{13-25}$ decreased from 50% to about 10% during 2013 to 2015, and then increased to reach around 60% in 2019.

Theil-Sen regression was used to quantitatively evaluate the changing trends of size-resolved PNCs, and they exhibited downward trends for all clusters during 2012-2015, especially in spring. The annual relative slopes of spring PNCs varied from -54.7% for Cluster F to -17.2% for Cluster B, -42.6% for Cluster A to -14.1% for Cluster B, and -40.7% for Cluster A to -17.5% for Cluster B per year for 13-25, 25-100, and 100-800 nm size ranges. The UFPs number was increased and the amplitude was greater during 2016-2019. The annual relative slopes of $N_{13-25}$ varied between 5.1% (fall) and 314.4% (winter), 8.0 (fall) and 135.5% (spring), 11.3% (fall) and 184.3% (winter), -4.5 (fall) and 59.1% (summer), 6.3% (fall) and 30.3% (spring), and 3.6% (fall) and 15.7% (spring) for Clusters A-F. The increased daytime net radiation, higher ambient temperature and lower relative humidity at noon for NPF events also could partly explain the higher $N_{13-25}$ induced by the more frequent nucleation events after 2016, especially in spring. The air mass history had a significant impact on urban PNSDs. The back trajectories were mainly from the adjacent regions of urban Lanzhou and less affected by long-range transport for NPF events, and the thus particles were not easily grown by coagulation during transport processes, which was helpful for occurrence of NPF events. In this study, the measurement campaign was conducted at a Chinese cities in west China, but the similar PNCs trends and influencing factors should be expected in other Chinese cities. In future work, we will established the PNSD observation network in some megacities to better evaluate the response of PNCs to emission mitigation policies in China.

*Author contributions.* Suping Zhao and Ye Yu designed the study. Suping Zhao analyzed the data with help from Ye Yu and Dahe Qin. Daiying Yin and Longxiang Dong collected and analyzed particle number size distributions and meteorology data during the campaign. Jianglin Li conducted the field experiment.

*Competing interests.* The authors declare that they have no conflict of interest.

*Acknowledgement.* The study is supported by National Natural Science Foundation of China (42075185; 41605103), Youth Innovation Promotion Association, CAS (2017462), Gansu Science and Technology Program



(20JR10RA037; 18JR2RA005), CAS "Light of West China" Program, and the Excellent Post-Doctoral Program (2016LH0020).

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

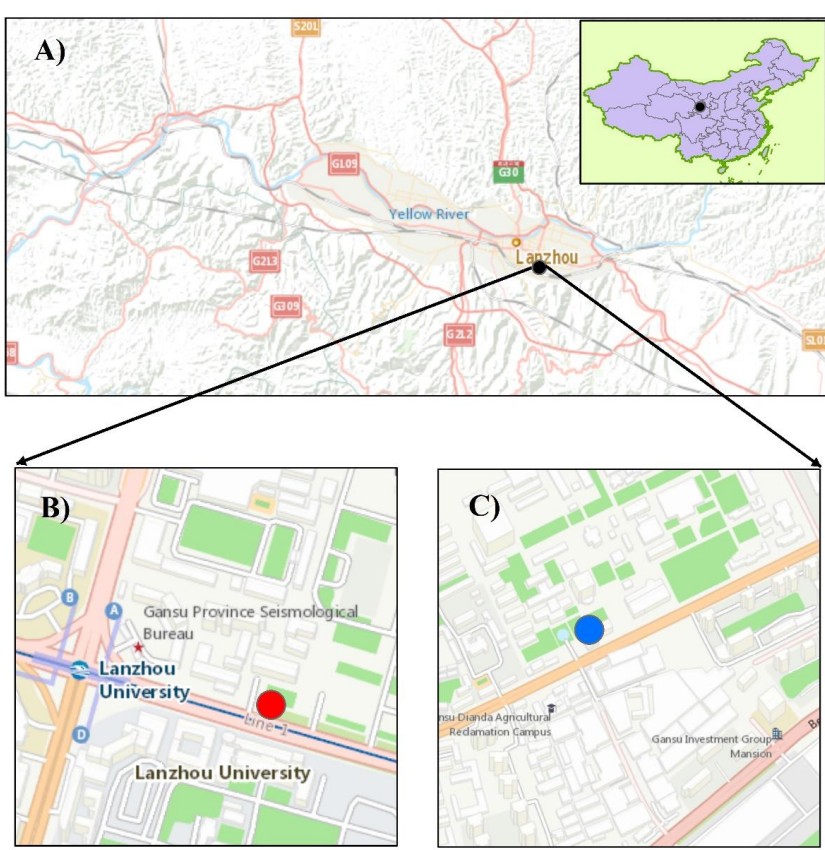

**Figure 1: Map of A) Lanzhou City, B) the sampling site (red dot) and C) reference station that measured PM$_{2.5}$ and O$_3$ (blue dot). The map is a pure reproduction of Google Maps where our own contribution is rather small, and we only added a few marks for our study locations.**

5    **Copyright © Google Maps.**

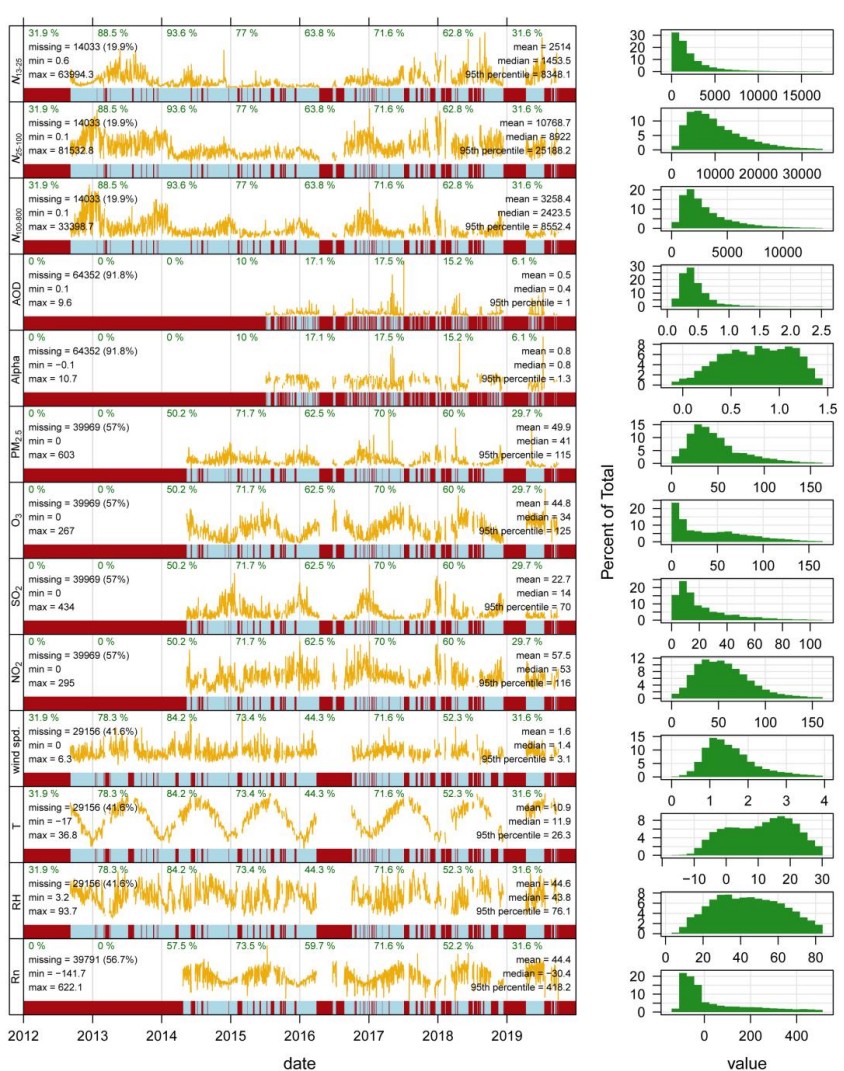

**Figure 2: Time series of daily average particle number in the three modes ($N_{13-25}$, $N_{25-100}$ and $N_{100-800}$), aerosol optical properties (AOD, Alpha), the criteria air pollutants (PM$_{2.5}$, O$_3$, SO$_2$, NO$_2$), and basic meteorological parameters and the corresponding probability density functions in urban Lanzhou during the campaign. The frequencies of missing values and the statistics are marked in each subplot. T, RH and Rn represent temperature, relative humidity and net radiation, respectively.**





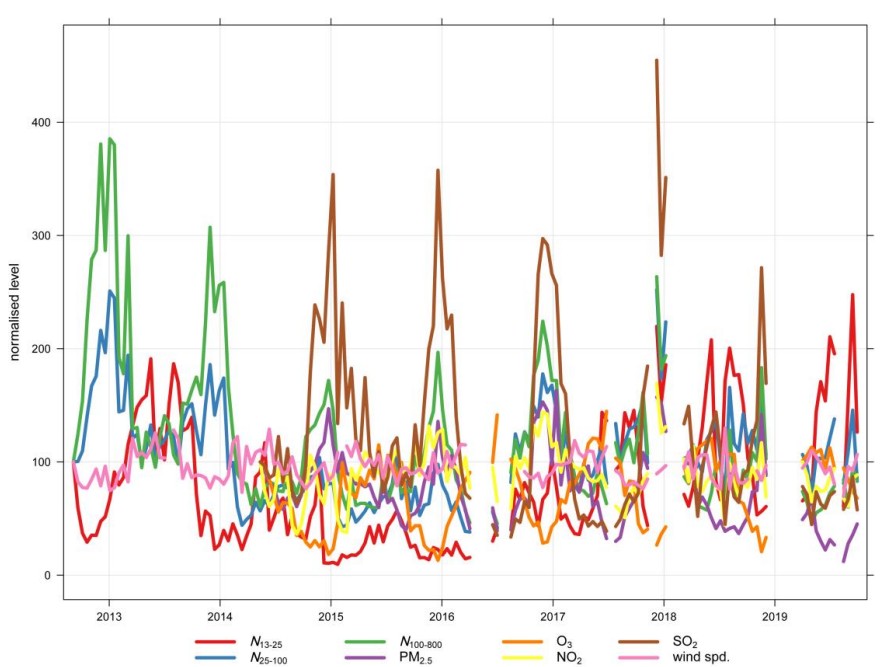

**Figure 3: Normalizing time series of monthly averaged data ($N_{13-25}$, $N_{25-100}$, $N_{100-800}$, PM$_{2.5}$, O$_3$, NO$_2$, SO$_2$, wind speed) to fix values to equal 100 at the beginning of September 2012.**

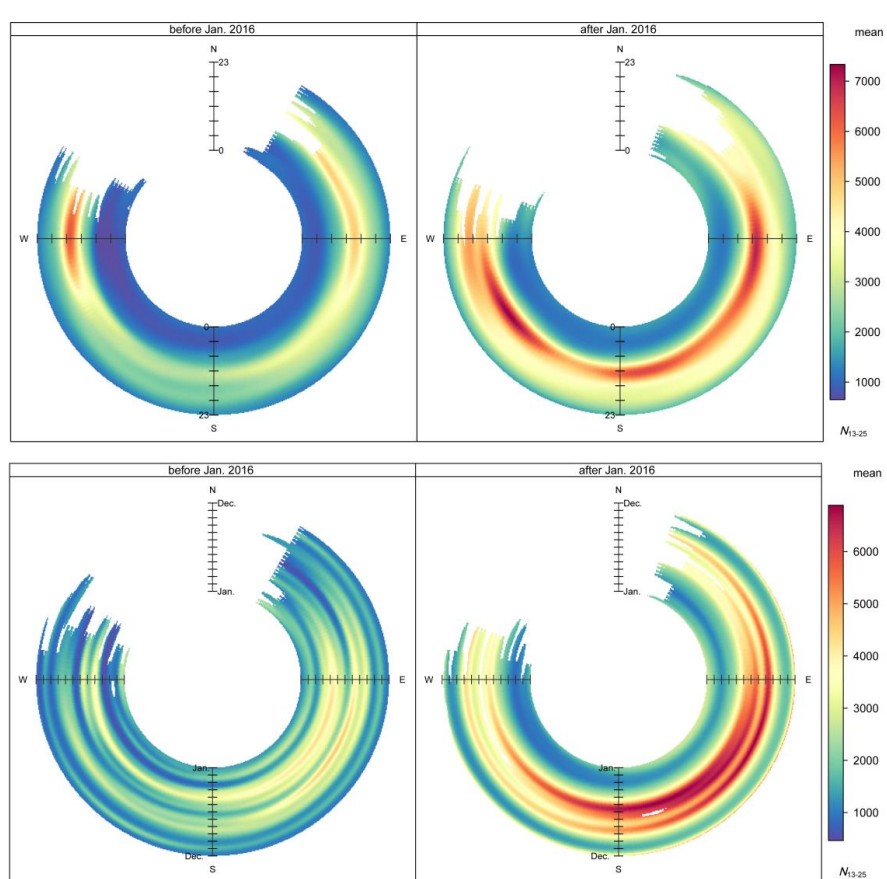

**Figure 4: Mean diurnal (upper panel) and annual (lower panel) variations of particle number in 13-25 size bin ($N_{13-25}$) as wind directions before and after January 2016.**



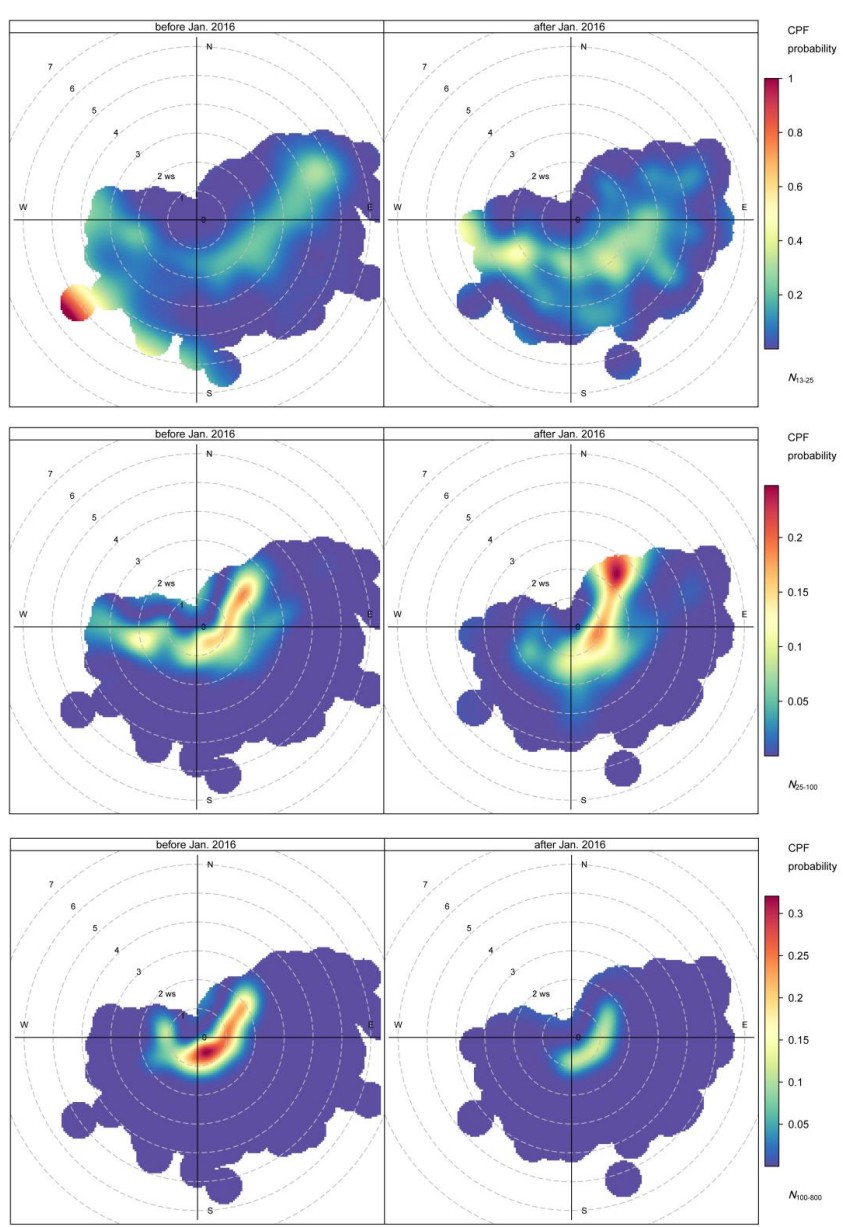

**Figure 5: Polar plot of $N_{13-25}$ (upper panel), $N_{25-100}$ (middle panel) and $N_{100-800}$ (lower panel) based on the CPF function before and after January 2016.**

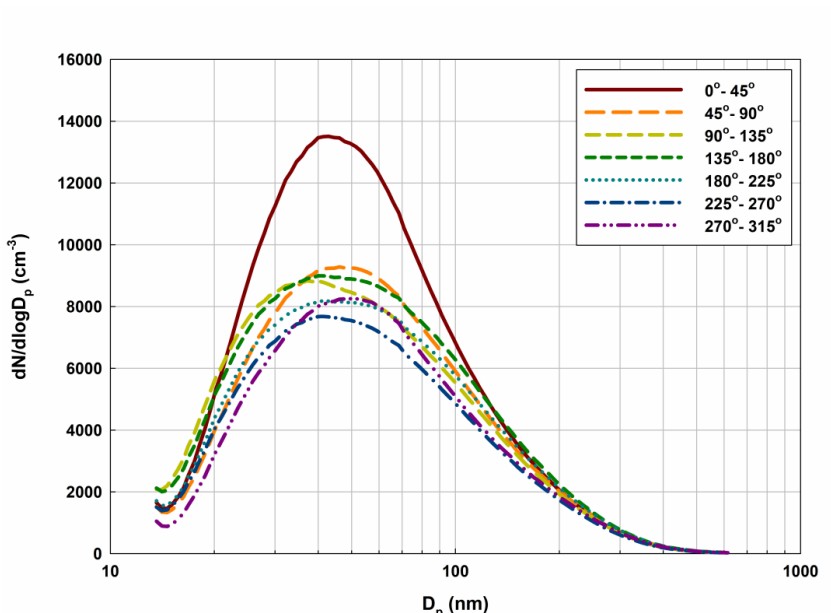

**Figure 6: Mean particle number size distributions by each sector of wind directions with interval of 45 degree.**

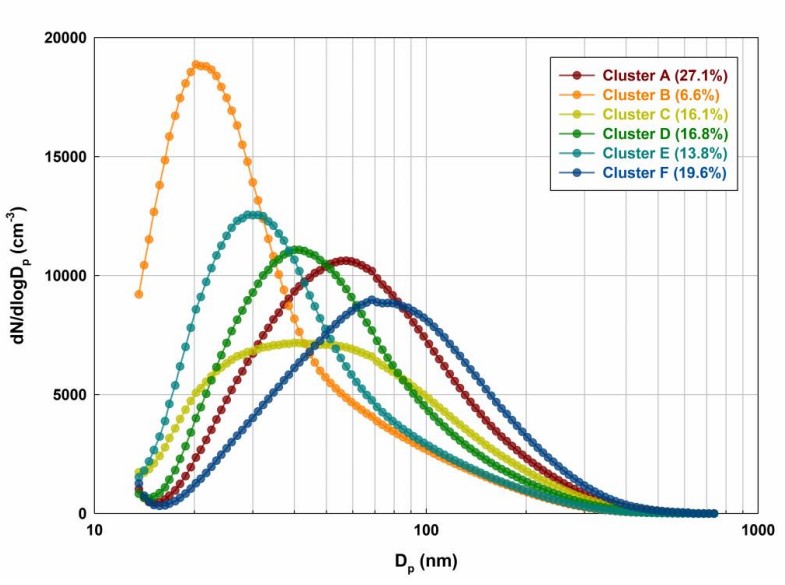





**Figure 7: Mean particle number size distribution for each typical cluster obtained by K-means clustering method. The occurrence frequencies of Clusters A-F were calculated during 2012-2019.**

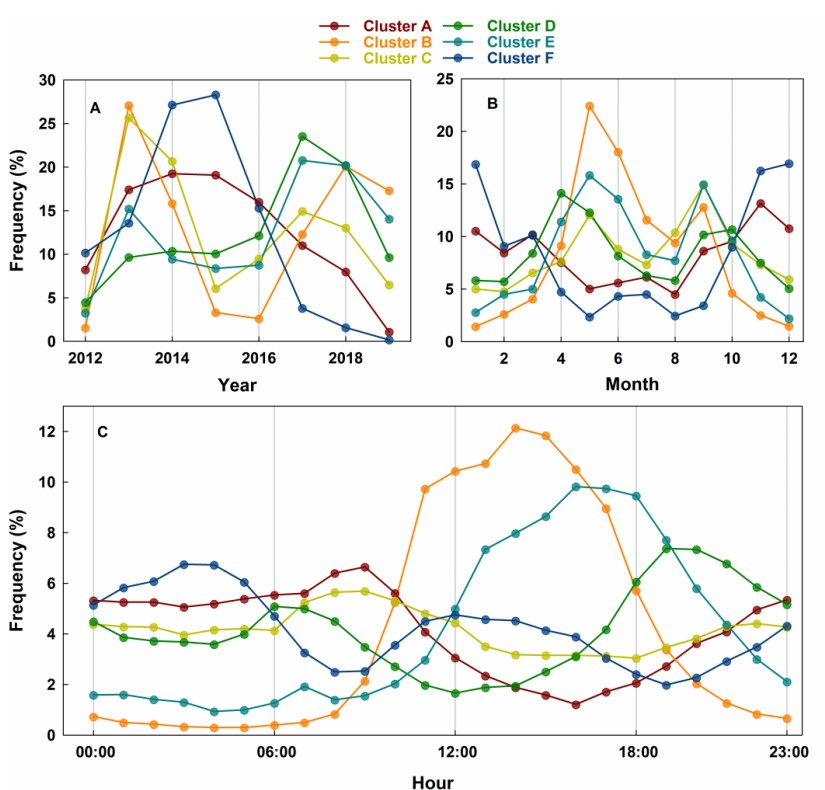

**Figure 8: (A) Inter-annual, (B) averagely annual and (C) diurnal variations of occurrence**

5      **frequencies for Clusters A-F during the measurement campaign.**





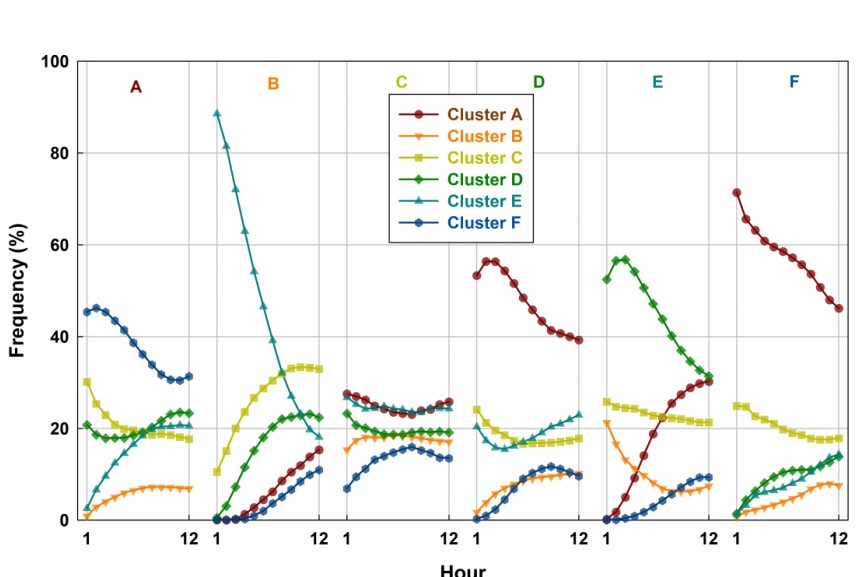

**Figure 9: Occurrence frequency of Clusters A-F during 1-12 hours after each cluster appeared during the entire measurement campaign.**

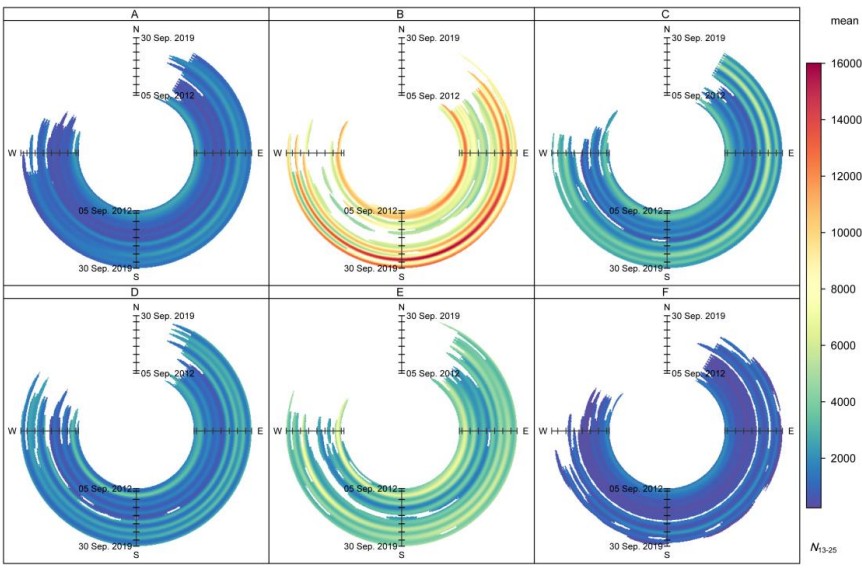

5   **Figure 10: Trends of daily mean number of particles in nucleation mode ($N_{13-25}$) as wind directions for each cluster during the entire measurement campaign.**

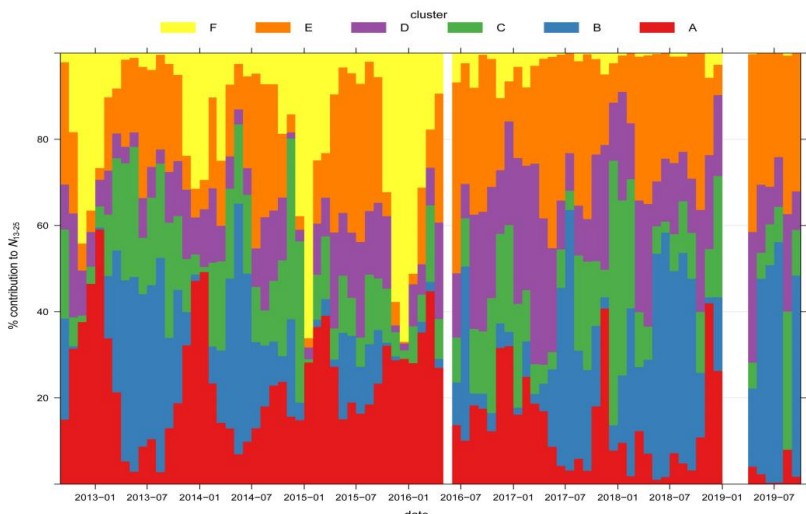

**Figure 11: Variations of contribution of each cluster to monthly averaged $N_{13-25}$ during the entire measurement campaign.**

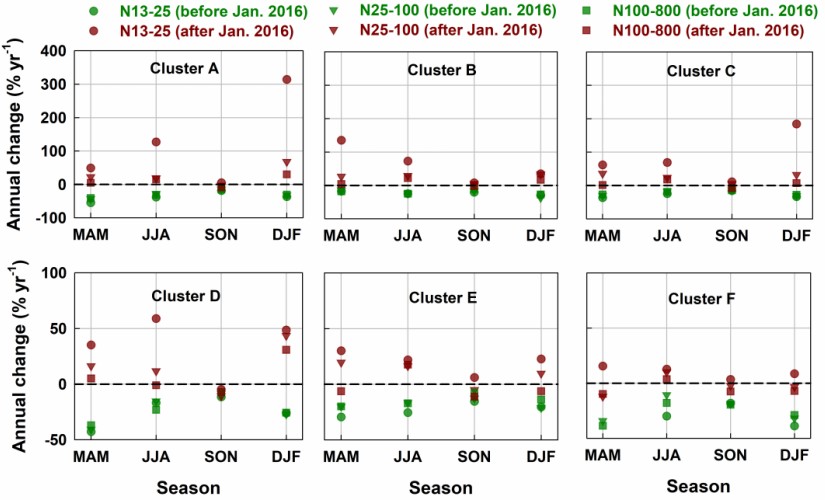

5    **Figure 12: Seasonal variations in the trends of PNCs in three modes before and after January 2016 for each cluster. The annual change is calculated by Theil-Sen regression. MAM, JJA, SON and DJF are spring, summer, fall and winter, respectively.**





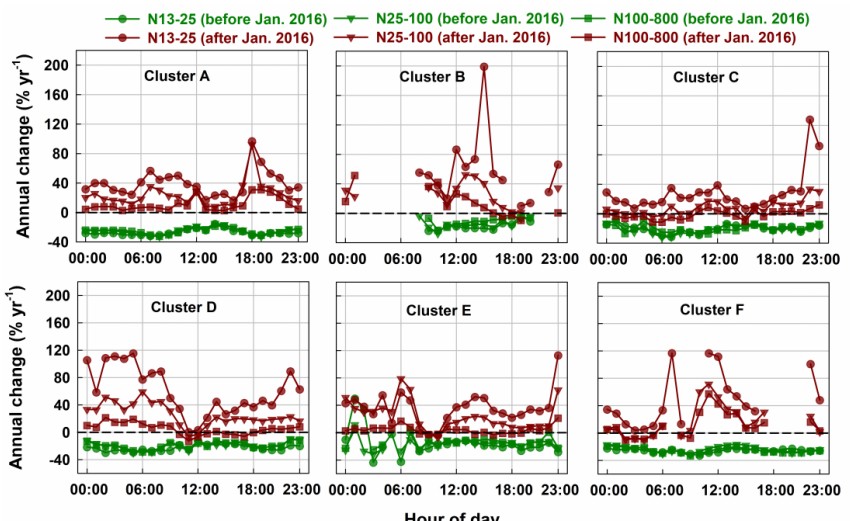

Figure 13: It is the same as Figure 12 but for diurnal variations.



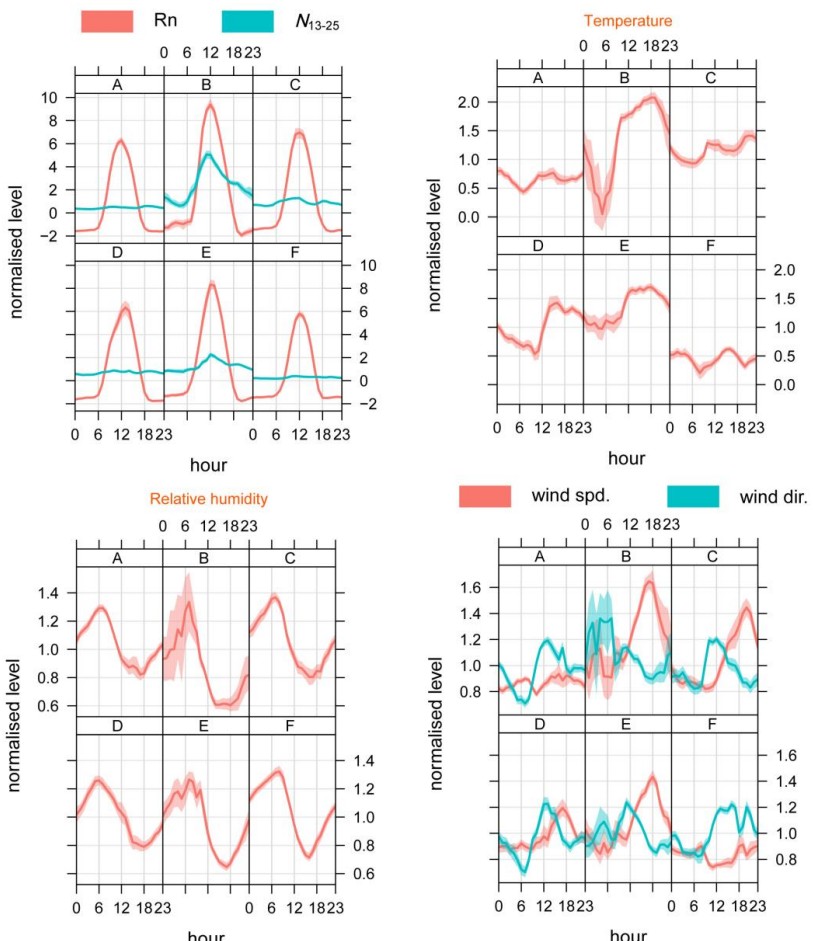

**Figure 14: Normalized diurnal variations of net radiation (Rn), $N_{13-25}$, temperature, relative humidity, and wind speed and direction for each cluster. The shading shows the estimated 95% confidence intervals.**





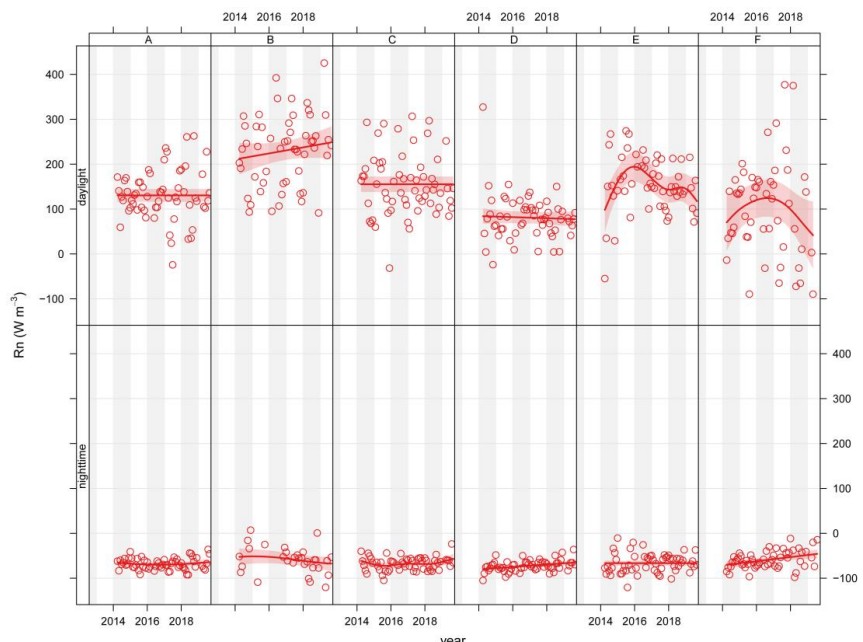

**Figure 15: Inter-annual variations of monthly average daylight and nighttime Rn for each cluster during the campaign. The smooth line is essentially determined using Generalized Additive Model, and the shading shows the estimated 95% confidence intervals.**

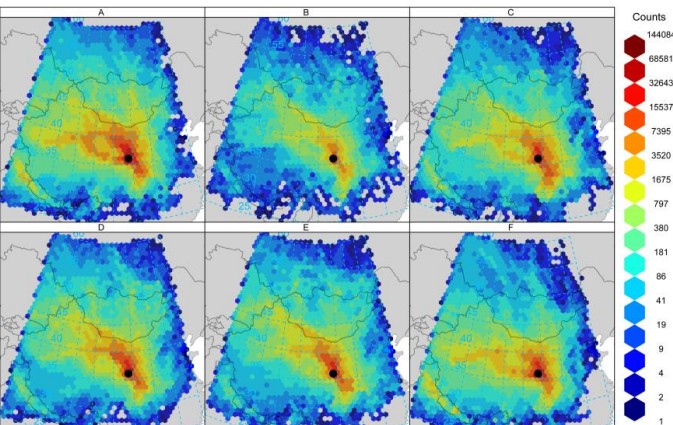

**Figure 16: Gridded back trajectory frequencies with hexagonal binning for each cluster. The five-day (120 h) trajectories were initialized at 500 m AGL. The black dot shows the geographic location of urban Lanzhou.**





**Table 1. Overview of experimentally determined particle number concentrations in the troposphere around the world. The duration of the measurement campaign was at least longer than 1 year (12 months).**

| Continent | Diameter, city, site and period | Number concentrations | | | Reference |
|---|---|---|---|---|---|
| Asia | Diameter range (nm) | 13-25 | 25-100 | 100-800 | This work |
| | Lanzhou, urban, 7.5 years | 2514 | 10769 | 3258 | |
| | Diameter range (nm) | 3-20 | 20-100 | 100-800 | Wang et al. (2013) |
| | Beijing, urban, 3 years | 5000 | 12300 | 6400 | |
| | Diameter range (nm) | 12-21 | 21-95 | 95-570 | Kivekas et al. (2009) |
| | Waliguan, remote rural, 22 months | 570 | 1060 | 430 | |
| | Diameter range (nm) | 3-25 | 25-100 | 100-1000 | Shen et al. (2011) |
| | Shangdianzhi, rural, 1.5 years | 3610 | 4430 | 3470 | |
| | Diameter range (nm) | | 20-100 | 100-685 | Kanawade et al. |
| | Kanpur, urban, 4 years | | 12400 | 18900 | (2014) |
| | Diameter range (nm) | 12-25 | 25-100 | 100-560 | Gani et al. (2020) |
| | Delhi, urban, 1.25 years | 8940 | 21690 | 11690 | |
| Europe | Diameter range (nm) | 8-25 | 25-90 | 90-460 | Dal Maso et al. (2008) |
| | Varrio, rural, 3 years | 143 | 429 | 304 | |
| | Diameter range (nm) | 8-30 | 30-100 | 100-700 | von Bismarck-Osten |
| | Copenhagen, rural, 3 years | 770 | 1813 | 751 | et al. (2013) |
| | Diameter range (nm) | 8-30 | 30-100 | 100-700 | von Bismarck-Osten |
| | Leipzig, roadside, 3 years | 5692 | 4962 | 2242 | et al. (2013) |
| | Diameter range (nm) | 8-30 | 30-100 | 100-700 | von Bismarck-Osten |
| | Helsinki, urban background, 3 years | 3080 | 3099 | 1053 | et al. (2013) |
| | Diameter range (nm) | 8-30 | 30-100 | 100-700 | von Bismarck-Osten |
| | London, urban background, 3 years | 1632 | 3825 | 1437 | et al. (2013) |
| North America | Diameter range (nm) | 10-50 | 50-100 | 100-500 | Wang et al. (2011) |
| | Rochester, urban, 8 years | 4730 | 1838 | 1073 | |
| | Diameter range (nm) | 3-20 | 20-100 | 100-1000 | Stanier et al. (2004) |
| | Pittsburgh, urban, 1 year | 9700 | 10100 | 2188 | |



**Table 2. Mean values of particle number in the three modes ($N_{13-25}$, $N_{25-100}$, and $N_{100-800}$), AOD, the concentrations of six criteria air pollutants (PM$_{2.5}$, PM$_{10}$, O$_3$, SO$_2$, NO$_2$, and CO) for each Cluster.**

| Cluster | $N_{13-25}$ | $N_{25-100}$ | $N_{100-800}$ | AOD | PM$_{2.5}$ | PM$_{10}$ | O$_3$ | SO$_2$ | NO$_2$ | CO |
|---|---|---|---|---|---|---|---|---|---|---|
| Units | cm$^{-3}$ | cm$^{-3}$ | cm$^{-3}$ | — | µg m$^{-3}$ | µg m$^{-3}$ | µg m$^{-3}$ | µg m$^{-3}$ | µg m$^{-3}$ | mg m$^{-3}$ |
| A | 1263.2 | 12156.5 | 3973.9 | 0.54 | 54.85 | 135.73 | 25.98 | 26.91 | 64.57 | 2.91 |
| B | 10370.4 | 9969.8 | 1504.7 | 0.39 | 31.42 | 86.53 | 92.77 | 13.56 | 40.67 | 2.73 |
| C | 2616.7 | 9071.8 | 2890.5 | 0.49 | 48.89 | 116.10 | 41.37 | 21.42 | 55.12 | 2.65 |
| D | 2010.0 | 11931.4 | 2301.6 | 0.55 | 43.92 | 124.13 | 46.28 | 16.85 | 57.68 | 2.33 |
| E | 4245.2 | 10806.9 | 1592.6 | 0.45 | 35.26 | 106.31 | 82.86 | 12.60 | 44.44 | 1.71 |
| F | 757.2 | 9492.2 | 5139.3 | 0.60 | 71.24 | 130.98 | 24.14 | 35.95 | 66.07 | 2.98 |

**Table 3. Mean values of meteorological parameters (wind speed, temperature, relative humidity and net radiation) and geometric median diameters (GMD$_{nuc}$, GMD$_{Ait}$ and GMD$_{acc}$ are for nucleation, Aitken and accumulation modes) fitted by Eq. (1) for each Cluster. WS, T, RH, and Rn are the abbreviations of wind speed, temperature, relative humidity and net radiation, respectively.**

| Cluster | WS | T | RH | Rn | GMD$_{nuc}$ | GMD$_{Ait}$ | GMD$_{acc}$ |
|---|---|---|---|---|---|---|---|
| Units | m s$^{-1}$ | ºC | % | W m$^{-2}$ | nm | nm | nm |
| A | 1.38 | 7.35 | 48.39 | 21.56 | 26.37 | 54.91 | 133.08 |
| B | 2.18 | 20.77 | 31.59 | 223.55 | 19.72 | 38.06 | 131.70 |
| C | 1.64 | 13.18 | 48.20 | 48.25 | 22.56 | 49.91 | 137.21 |
| D | 1.57 | 11.73 | 44.31 | -3.03 | 25.73 | 44.33 | 128.89 |
| E | 1.94 | 17.47 | 36.19 | 100.16 | 23.18 | 38.74 | 122.96 |
| F | 1.34 | 5.34 | 47.03 | 12.82 | 25.69 | 62.35 | 136.55 |