# Peer review of "Response of particle number concentrations to Clean Air Action: Lessons from the first long-term aerosol measurements in a typical urban valley, West China"

_Atmospheric Chemistry and Physics, 2021_

## Author Comment (AC2)

**RESPONDS TO COMMENTS FROM REFEREE #1**

First of all, we appreciate your very positive evaluation of our work. The responses of your specific comments/questions are outlined in detail below.

**Specific comments:**

(1) The analysis of long-term analysis of NPF and its frequency is incomplete without calculating and discussing the changes in the condensation (and coagulation) sink during this period — I highly recommend including this analysis. Furthermore, the authors could also calculate H2SO4 proxy (Dada et al., 2020) to estimate the changes in the NPF precursors over the study period.

**Response:** Thank you for your constructive suggestions and providing important reference. As you said, the condensation (and coagulation) sink and the precursors were very important for analyzing NPF and its frequency, which should be included in this study.

Referred to the methods presented in Dal Maso et al. (2005), NPF events were identified for a day. Number concentration sharply increased in the nucleation mode size range (13-25 nm) and prevailed for at least an hour. Additionally, the particle size was required to increase during the next few hours. The parameters describing NPF events such as formation and growth rates (*J*D, *GR* and hereafter), condensation and coagulation sink (*CS*, *CoagS* and hereafter) were calculated in this study. *GR* can be calculated with the time evolution of geometric mean diameter (*GMD*) of the nucleation mode obtained by parameterizing PNSD, and it can be expressed as

$$GR = \frac{dGMD}{dt}$$
(1)

The formation rates  $(J_D)$  can be calculated by the below equation:

$$J_D = \frac{dN_{nuc}}{dt} + F_{coag}$$
(2)

where the first term in the right hand side  $(dN_{nuc}/dt)$  represents the observed change of in number concentration of newly formed particles (Zhao et al., 2021). The second term is the loss of newly formed particles induced by coagulation scavenging, and can be obtained with the below equation:

$$F_{coag} = CoagS_{nuc}N_{nuc}$$
(3)

Coagulation sink of nucleation mode particles (CoagSnuc) is defined as

$$CoagS(D_p) = \int K(D_p, D_p) n(D_p) dD_p$$
(4)

where  $K(D'_{\rho}, D_{\rho})$  is the coagulation coefficient of particles with sizes of  $D_{\rho}$  and  $D'_{\rho}$ , calculated by the method of Fuchs (1964). The reference size  $(D_{\rho})$  is assumed to be

the *GMD* of the nucleation mode. An average  $CoagS_{nuc}N_{nuc}$  over each formation period was taken during the campaign.

The condensation sink (CS) can be expressed as

$$CS = 2\pi D \sum \beta_m \left( D_{p,i} \right) D_{p,i} N_i \tag{5}$$

where  $D_{p,i}$  and  $N_i$  are particle diameter and the corresponding number concentration in size class *i*. D is the diffusion coefficient of the condensing vapor, usually assumed to be sulfuric acid.  $\theta_m$  represents a transition-regime correction (Kulmala et al., 2012),

$$\beta_m = \frac{1 + Kn}{1 + 1.677Kn + 1.333Kn^2} \tag{6}$$

defined as a function of the Knudsen number,  $Kn = 2\lambda/D_{p,i}$ . Furthermore, based on the method presented in Dada et al. (2020), H2SO4 proxy was calculated to estimate the changes in the NPF precursors over the study period, and the equation was given as follows.

$$\left[H_{2}SO_{4}\right] = -\frac{CS}{2 \cdot \left(9.9 \times 10^{-9}\right)} + \left[\left(\frac{CS}{2 \cdot \left(9.9 \times 10^{-9}\right)}\right)^{2} + \frac{\left[SO_{2}\right]}{\left(9.9 \times 10^{-9}\right)}\left(1.6 \times 10^{-9} \cdot \text{GlobRad}\right)\right]^{1/2}$$
(7)

where *CS* was calculated by Equation (6). SO2 concentrations are measured by the ultraviolet fluorescence method, and Global radiation (GlobRad) was measured by an SMP3 pyranometer (Kipp and Zonen, the Netherlands) during the campaign. In addition, the peak sizes of PNSDs are determined as mode diameters.

To better analyze long-term trend of NPF event and the relevant parameters during 2012-2019, Figure 11 illustrates the inter-annual statistics of the trends of NPF frequency, mode diameter, and formation and growth rates. Furthermore, condensation and coagulation sink (CS, CoagS) and H2SO4 proxy were also calculated over the study period. Similar with the opposite  $N_{13-25}$  trend between the two contrasting periods (Figures 8-9), the occurrence frequency of NPF events decreased from ~ 30% to less than 5% until 2016 and then increased to more than 30% in 2019. The particle has been becoming much finer since 2015 due to more frequent NPF events (Figure 11B). The temporal variations of PNCs in nucleation mode ( $dN_{nuc}/dt$ , Figure 11C) and coagulation scavenging effect (Fcoag, Figure 11D) followed similar inter-annual variations of NPF frequency. The contribution of coagulation loss flux  $F_{coag}$  to total observed rate was on average 37%, which was close to the average ratio of coagulation loss to formation rate in urban Beijing, 0.41 (Yue et al., 2010), suggesting that coagulation loss was the same important as  $dN_{nuc}/dt$ . The formation rate  $(J_D)$  ranged from 0.2 to 16.2 cm-3 s-1 in urban Lanzhou, which was lower than the observations at some urban sites, such as in Beijing, 3.3-81.4 cm-3 s-1 (Wu et al.,

2007), St. Louis, with the mean value of 17.0 cm-3 s-1 (Qian et al., 2007), but much higher than that in regional nucleation episodes 0.01-10 cm-3 s-1 at the most other sites (Kulmala et al., 2004).

Compared with  $J_D$ , *GR* varied less in inter-annual scale, and ranged from 0.5 to 14.9 nm h-1, slightly higher than that in urban Beijing, 0.3-11.2 nm h-1 (Wu et al., 2007), and also within the range of typical particle growth rate 1-20 nm h-1 in mid-latitudes (Kulmala et al., 2004). The inter-annual variation of condensation sink (*CS*) was consistent with that of NPF frequent and formation rate with the range between  $7.3 \times 10^{-4}$  and  $5.1 \times 10^{-2}$  s-1 with mean value of  $1.4 \times 10^{-2}$  s-1 (Figure 11F), which was comparable with the calculated value of 0.02 s-1 during NPF events in the North China Plain (Shen et al., 2011). Based on the method presented in Dada et al. (2020), we also calculated H2SO4 proxy to estimate the changes in the NPF precursors over the study period (Figure 11G). The H2SO4 proxy varied from  $3.3 \times 10^7$  to  $6.0 \times 10^8$  cm-3 with average concentration of  $2.5 \times 10^8$  cm-3 over the study period, which was slightly higher than that in urban Beijing (Dada et al., 2020) due to more coal combustion and basin terrain in urban Lanzhou. The used methods and the corresponding analyses and discussion will be included in the revised version of our manuscript.

---

## Author Response (AR1)

**RESPONDS TO COMMENTS FROM REFEREE #1**

First of all, we appreciate your very positive evaluation of our work. The responses of your specific comments/questions are outlined in detail below.

**Specific comments:**

(1) The analysis of long-term analysis of NPF and its frequency is incomplete without calculating and discussing the changes in the condensation (and coagulation) sink during this period — I highly recommend including this analysis. Furthermore, the authors could also calculate H2SO4 proxy (Dada et al., 2020) to estimate the changes in the NPF precursors over the study period.

**Response:** Thank you for your constructive suggestions and providing important reference. As you said, the condensation (and coagulation) sink and the precursors were very important for analyzing NPF and its frequency, which should be included in this study.

[revised manuscript text omitted]

(2) The authors separate the study into two contrasting periods (before and after Jan 2016). However, based on the timeseries in Figure 2, it seems that 2013 is an unusually polluted year even for 2012–2015 period. I think presenting the average particle size distribution surface plots for each year (by season) can be instructive for highlighting the overall similarities and differences of each year of the study period (perhaps in the SI).

**Response:** Thank you for catching that. As you said, the average particle number size distribution (PNSD) surface plots in four seasons for each year during the campaign are presented in Figures S1-S4 to highlight the overall similarities and differences of each year during the study period. The mode diameter of PNSD shifts to smaller particle size in four seasons from 2012 to 2019. Particle number in Aitken and accumulation modes declined largely in autumn and winter during the study periods maybe due to the even strictest emission control policies in recent years. However, in spring and summer, the nucleation mode particle number increased significantly after 2016, which can be partly modulated by NPF events. Figures S1-S4 and the above corresponding discussions will be added to the revised version of our manuscript.

[Figure]

Figure S1: Evolutions of hourly average particle number size distributions (PNSD) in spring for each year during the campaign. The white gaps in the subplots represent missing data due to failures or routine maintenance of the instruments.

[Figure]

Figure S2: Evolutions of hourly average particle number size distributions (PNSD) in summer for each year during the campaign. The white gaps in the subplots represent missing data due to failures or routine maintenance of the instruments.

[Figure]

Figure S3: Evolutions of hourly average particle number size distributions (PNSD) in autumn for each year during the campaign. The white gaps in the subplots represent missing data due to failures or routine maintenance of the instruments.

[Figure]

Figure S4: Evolutions of hourly average particle number size distributions (PNSD) in winter for each year during the campaign. The white gaps in the subplots represent missing data due to failures or routine maintenance of the instruments.

(3) Many of the graphics in the manuscript are well made and explain the central themes of the study well (e.g., I think Figure 4 is excellent). However, I think the readability of manuscript can be improved if some of the graphics that are supplementary to the analysis are removed or moved to the SI (e.g., Figure 3). Furthermore, some more details should be provided in the captions of the figures as they should be interpretable independent of the text (e.g., Figure 9 was not clear to me).

**Response:** Thank you for your suggestions. To improve the readability of manuscript, Figures 3, 5, 11, 15 and 16 that are supplementary to the analysis will be moved to the SI. Furthermore, some more details will be provided in the captions of the all figures.

(4) Page 9, Line 4 ("The nucleation mode particles also can grow…"). I do not think this sentence is correct or required for this discussion. The timescales for such a transport (from ground to the sampling inlet) is likely to be much shorter than for the particles to grow from nucleation to Aitken mode.

**Response:** Thank you for your reminder. The sentence in Line 4 of Page 9 will be deleted in the revised version of our manuscript.

(5) Page 17, Line 15 ("NPF events predominantly occurred under dry and sunny weather conditions"). This should be discussed with more nuance based on more recent literature. For example, according to a relatively recent review paper on NPF, "The observed factors that favor the occurrence of regional NPF include a high intensity of solar radiation, low RH, high gas-phase sulfuric acid concentration, and low pre-existing particle loading, i.e. low CS and CoagS" (Kerminen et al., 2018).

**Response:** Thank you for your good suggestions. According to a relatively recent review on regional NPF in different environments of the global troposphere, the observed factors that favor the occurrence of regional NPF include a high intensity of solar radiation, low RH, high gas-phase sulfuric acid concentration, and low pre-existing aerosol loading (Kerminen et al., 2018). The possible reasons for the apparently close connection between the ambient RH and occurrence of NPF have been proposed, including the typically negative feedback of high RH on the solar radiation intensity, photochemical reactions and atmospheric lifetime of aerosol precursor vapors. The effect of the ambient temperature ($T$) on NPF shows very different responses between different studies, which is probably related to the simultaneous presence of several temperature-dependent processes that may either enhance or suppress NPF. Therefore, the meteorological parameters affect NPF process by modulating the condensation and coagulation sink. We will add the above discussion to the revised manuscript.

(6) For figures in manuscript or SI: (i) Include units of all parameters (where applicable); (ii) Avoid using captions such as "Same as Figure X, but for…"; (iii) Use continuous colorbars when using surface plots (Fig. 17 and Fig. S7).

**Response:** Thank you for your good suggestions. The all figures and the corresponding captions in new manuscript and SI will be revised according to your three suggestions.

---

## Author Response (AR2)

**RESPONDS TO COMMENTS FROM REFEREE #1**

First of all, we appreciate your very positive evaluation of our work. The responses of your specific comments/questions are outlined in detail below.

**Specific comments:**

(1) I am a little surprised by Figure 11 B and F. I would not have expected the condensation sink to increase over the last ~5 years even as the mode diameter decreases over this period. It will be helpful to have the barplots for N13-25, N25-100, and N100-800 in this figure.

**Response:** Thank you for catching that. As you said, low condensation sink and the meteorological factors like low RH and high solar radiation favor the NPF events (Boy and Kulmala, 2012). We updated the $CS$ data and non-NPF events also were included in the revised Figure 11F, and $N_{13-25}$, $N_{25-100}$ and $N_{100-800}$ variations also were given in the figure. As it can be seen from Figure 11F, the mean condensation sink was between 0 and 0.01 s$^{-1}$ with less fluctuation during the campaign. Shen et al. (2011) found that mean value of $CS$ was 0.02 s$^{-1}$ during NPF events in North China Plain, which was much higher than our results. Therefore, NPF events was less impacted by condensation sink during our campaign. The less varied $N_{100-800}$ during 2015-2019 as compared to that during 2012-2014 may be related to condensation of low volatile vapors which resulted in relatively high condensation sink. The above discussion and the revised Figure 11 will be added to the revised version of our manuscript.

[Figure]

Figure 11: Inter-annual statistics of the trends of NPF frequency, mode diameter, formation ($dN_{nuc}/dt$, $F_{coag}$) and growth rates, $CS$ and $H_2SO_4$ proxy and number concentrations in the three bins ($N_{13-25}$, $N_{25-100}$, $N_{100-800}$) during the campaign. The lines inside the box denotes the median slope, the two whiskers and the top and bottom of the box denote the 5th, 95th, 75th and 25th percentiles.

(2) The condensation sink values and discussion should be incorporated in the cluster analysis (specifically in Table 2).

**Response:** Thank you for your suggestions, which will largely improve the manuscript. The condensation sink values were incorporated in the cluster analysis (see also the below Table 2). The condensation sink ranged from $2.12×10^{-3}$ $s^{-1}$ for Cluster F to $1.38×10^{-2}$ $s^{-1}$ for Cluster B during the campaign. The $CS$ values for Clusters B and F, representing new particle formation and growth events, were much higher than that for other clusters, but they was even lower than $CS$ during NPF events in North China Plain ($0.02$ $s^{-1}$, Shen et al., 2011). Therefore, the large PNSD discrepancy among the clusters may be less influenced by condensation sink during the measurement campaign. $CS$ may be not a key factor modulating occurrence of NPF events at urban Lanzhou in west China, and NPF was mainly affected by meteorological variables and coagulation effects. The above discussion and the revised Table 2 will be included in the revised version of our manuscript.

**Table 2. Mean values of particle number in the three modes ($N_{13-25}$, $N_{25-100}$, and $N_{100-800}$), AOD, the concentrations of six criteria air pollutants (PM$_{2.5}$, PM$_{10}$, O$_3$, SO$_2$, NO$_2$, and CO) and condensation sink ($CS$) for each Cluster.**

| Cluster | $N_{13-25}$ | $N_{25-100}$ | $N_{100-800}$ | AOD | PM$_{2.5}$ | PM$_{10}$ | O$_3$ | SO$_2$ | NO$_2$ | CO | $CS$ |
|---|---|---|---|---|---|---|---|---|---|---|---|
| Units | cm$^{-3}$ | cm$^{-3}$ | cm$^{-3}$ | — | µg m$^{-3}$ | µg m$^{-3}$ | µg m$^{-3}$ | µg m$^{-3}$ | µg m$^{-3}$ | mg m$^{-3}$ | $10^{-3}$ s$^{-1}$ |
| A | 1263.2 | 12156.5 | 3973.9 | 0.54 | 54.85 | 135.73 | 25.98 | 26.91 | 64.57 | 2.91 | 3.64 |
| B | 10370.4 | 9969.8 | 1504.7 | 0.39 | 31.42 | 86.53 | 92.77 | 13.56 | 40.67 | 2.73 | 13.8 |
| C | 2616.7 | 9071.8 | 2890.5 | 0.49 | 48.89 | 116.10 | 41.37 | 21.42 | 55.12 | 2.65 | 4.42 |
| D | 2010.0 | 11931.4 | 2301.6 | 0.55 | 43.92 | 124.13 | 46.28 | 16.85 | 57.68 | 2.33 | 5.33 |
| E | 4245.2 | 10806.9 | 1592.6 | 0.45 | 35.26 | 106.31 | 82.86 | 12.60 | 44.44 | 1.71 | 7.88 |
| F | 757.2 | 9492.2 | 5139.3 | 0.60 | 71.24 | 130.98 | 24.14 | 35.95 | 66.07 | 2.98 | 2.12 |

Reference

Boy, M. and Kulmala, M.: Nucleation events in the continental boundary layer: Influence of physical and meteorological parameters, Atmos. Chem. Phys., 2, 1–16, doi:10.5194/acp-2-1-2002, 2002.

Shen, X. J., Sun, J. Y., Zhang, Y. M., Wehner, B., Nowak, A., Tuch, T., Zhang, X. C., Wang, T. T., Zhou, H. G., Zhang, X. L., Dong, F., Birmili, W., and Wiedensohler, A.: First long-term study of particle number size distributions and new particle formation events of regional aerosol in the North China Plain, Atmospheric Chemistry and Physics, 11, 1565–1580, 2011.